# A Fitting Recognition Approach Combining Depth-Attention YOLOv5 and Prior Synthetic Dataset

**Jie Zhang** , **Jin Lei \*** , **Xinyan Qin, Bo Li, Zhaojun Li, Huidong Li, Yujie Zeng and Jie Song**

College of Mechanical and Electrical Engineering, Shihezi University, Shihezi 832003, China
* Correspondence: jinlei@shzu.edu.cn

**Abstract:** To address power transmission lines (PTLs) traveling through complex environments leading to misdetections and omissions in fitting recognition using cameras, we propose a fitting recognition approach combining depth-attention YOLOv5 and prior synthetic dataset to improve the validity of fitting recognition. First, datasets with inspection features are automatically synthesized based on prior series data, achieving better results with a smaller data volume for the deep learning model and reducing the cost of obtaining fitting datasets. Next, a unique data collection mode is proposed using a developed flying-walking power transmission line inspection robot (FPTLIR) as the acquisition platform. The obtained image data in this collection mode has obvious time-space, stability, and depth difference, fusing the two data types in the deep learning model to improve the accuracy. Finally, a depth-attention mechanism is proposed to change the attention on the images with depth information, reducing the probability of model misdetection and omission. Test field experiments results show that compared with YOLOv5, the mAP5095 (mean average precision on step size 0.05 for thresholds from 0.5 to 0.95) of our depth-attention YOLOv5 model for fitting is 68.1%, the recall is 98.3%, and the precision is 98.3%. Among them, AP, recall, and precision increased by 5.2%, 4.8%, and 4.1%, respectively. Test field experiments verify the feasibility of the depth-attention YOLOv5. Line field experiments results show that the mAP5095 of our depth-attention YOLOv5 model for fittings is 64.6%, and the mAPs of each class are improved compared with other attention mechanisms. The inference speed of depth-attention YOLOv5 is 3 ms slower than the standard YOLOv5 model and 10 ms to 15 ms faster than other attention mechanisms, verifying the validity of the depth-attention YOLOv5. The proposed approach improves the accuracy of the fitting recognition on PTLs, providing a recognition and localization basis for the automation and intelligence of inspection robots.

**Keywords:** deep learning; synthetic dataset; depth map; attention mechanism; fitting recognition

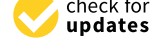



## 1. Introduction

Power transmission line (PTL) is the main component of electrical energy transmission, which is necessary for economic development and human life. Inspection of PTL is the most effective way to prevent some potential damages from power outages. PTLs are widely erected in harsh fields, which brings many great challenges for inspection. Manual inspection gradually cannot meet the needs of inspection of PTL system [1]. Meanwhile, the inspection robot is developing towards intelligence and automation; thus, it is a promising method to replace manual inspection and improve inspection efficiency [2].

At present, inspection robots for PTLs can be divided into three categories: walking inspection robots [3–5], flying inspection robots [6,7], and hybrid inspection robots [8]. Walking inspection robots walk along the PTL and collect more reliable data, but the robots have limitations in crossing obstacles and taking up and off PTL. Flying inspection robots are more flexible but do not obtain as high-quality inspection data as walking inspection robots. Hybrid inspection robots combine flying and walking inspection robots in their structures, which has both advantages and better application prospects. The inspection

methods of the robots include vision-based detection, thermal detection, radar-based detection, multi-sensor, etc. Among them, vision-based detection is the most convenient and economical. In addition, robots can be localized by vision-based detection, which is one of the key technologies for robot automation and intelligence.

In recent years, many methods of inspection fittings have been proposed in the vision field, such as image-based methods [9–11], shallow learning algorithms [12–14], and deep learning algorithms [15,16]. The image-based method does not require mass data with a good detection speed, but its robustness is not high. Shallow learning algorithms rely on effective feature design, so they are suitable for scenario-specific recognition. Nevertheless, the generalization of this method is not good. Deep learning methods have strong feature representation, better robustness, and generalization in complex environments, showing great potential in target recognition. Deep learning methods are divided into two categories: two-stage methods and one-stage methods. The two-stage methods have two key processes: extracting the candidate regions and recognizing the regions. Although this method is more accurate than the one-stage method, its disadvantages are obvious: each image requires many candidate regions and large duplicate information in adjacent regions, resulting in high computational effort and slow detection speed. The representative algorithms are RCNN series [17–20] and SPP-Net [21]. The core of the one-stage methods is the regression of the recognition results, which eliminates the process of extracting the candidate region of the two-stage method. Although the accuracy is lost, the detection speed is faster. The representative algorithm of the one-stage method is the YOLO series algorithm [22–28], which has been developed for six years since it was first proposed in 2016. However, it is still challenging to apply YOLO to recognize the fittings on PTLs. Firstly, there are many fitting recognitions, and the requirements of dataset volume are large. In addition, PTLs are mostly located in the harsh field, leading to the difficult collection of fitting datasets and the high consumption of human and financial resources. Secondly, complex environments and changing weather around PTLs make the background miscellaneous and inconsistent quality of the collected image data.

To address these challenges, we propose a fitting recognition approach combining depth-attention YOLOv5 and a prior synthetic dataset based on a developed flying-walking power transmission line inspection robot (FPTLIR). The FPTLIR collects data along the ground line from far and near, including RGB images and depth maps. According to the characteristics of the collected data, a synthetic dataset based on prior series data is synthesized to overcome the difficulties of dataset collection, greatly reducing the cost of producing the dataset and improving the efficiency of algorithm development. Depth-attention YOLOv5 is a recognition approach based on the YOLOv5 fusion depth-attention mechanism. The inputs for the training stage are RGB images, and the inputs for the detection stage are RGB images and matched depth maps from the FPTLIR's collection. The collected depth maps reflect the depth difference between the background and the target. Therefore, the proposed approach could improve attention on the target and reduce attention from the background. The depth-attention mechanism is introduced after the head layer of YOLOv5, enabling in the detection stage and freezing in the training stage. The depth-attention mechanism could also be directly added after the trained models. The main three contributions of this paper are as follows:

1.  Synthesizing automatic synthetic datasets with inspection features based on prior fitting series, prior inspection-view series, prior topography series, and prior time series. So, the proportion of inspection-unrelated features in the dataset is small, achieving better results with a smaller data volume for the deep learning model. In addition, the cost of obtained dataset acquisition is further reduced compared to other synthetic and real data collection methods.

2.  Proposing a unique data collection mode using FPTLIR as the acquisition platform, i.e., walking along the top ground line to collect sequence image data. The obtained image data in this collection mode has obvious time-space, stability, and depth difference, in particular, the RGB images and depth maps with depth differences between the

fitting and the background, fusing the two data types in the deep learning model to improve the model accuracy.

3. Proposing a depth-attention mechanism activated in the detection stage to change the attention on the images according to depth information. It is directly introduced into the deep learning model without secondary training, reducing the probability of model misdetection and omission more conveniently and quickly. The experiments prove the feasibility and validity of the proposed approach, and the results are significantly better than other attention mechanisms, which have practical application significance.

This paper is organized as follows: Section 2 presents the related work. Section 3 describes the FPTLR structure and characteristics of the collected data. Section 4 provides the rationale for our proposed approach. Section 5 conducts test field experiments and line field experiments to verify the feasibility and validity of depth-attention YOLOv5, respectively. Section 6 provides the discussion. Conclusions are drawn in the last section.

## 2. Related Works

### 2.1. Fitting Dataset

The methods of obtaining the fitting dataset include field collection [29], image stitching [30], and data synthesis [31]. Field collection methods can collect real images, but they are expensive and difficult to collect complete data. Image stitching can effectively increase the dataset volume of small samples, but the quality and quantity of the dataset are subject to manual operation. Data synthesis requires only tuning parameters to automatically generate a low-cost, large-volume dataset, providing a promising way to obtain the datasets. There are few public datasets of fittings, and the variety and volume of datasets are limited. It is difficult to obtain enough datasets to meet the requirements. Table 1 lists some public datasets on the Internet.

**Table 1.** Public dataset of fittings.

| Dataset | Brief Description | Quantity | Type | Website |
|---|---|---|---|---|
| Insulator [30] | Aerial images, including normal insulators and defective insulators (CPLID) | 848 | Real and Synthetic | https://github.com/InsulatorData/InsulatorDataSet (accessed on 30 October 2022) |
| Insulator [31] | Synthesis datasets based on prior series data using virtual models | 10,800 | Synthetic | https://github.com/zjlanthe/synthetic_data_insulator_blender (accessed on 30 October 2022) |
| Insulator [29] | Low- to medium-voltage porcelain insulators | 5939 | Real | https://ieee-dataport.org/open-access/low-medium-voltage-porcelain-insulators-primary-distribution-systems-dataset (accessed on 30 October 2022) |
| Insulator | The insulator defect image dataset (IDID) | 1596 | Real | https://ieee-dataport.org/competitions/insulator-defect-detection (accessed on 30 October 2022) |

CPLID [30] included 600 normal insulators and 248 defective insulators by unmanned aerial vehicle (UAV) capture, where the defective insulators were expanded using the image stitching method due to their small number. A synthetic insulator dataset [31] included 10,800 complete insulators annotated in VOC2007 format. This study provided a synthesis method for a fitting dataset, obtaining 98% of mean average precision (mAP) using the trained YOLOX model on a real test dataset. EhabUr Rahman provided primary distribution system datasets, including 5939 normal and damaged insulators photographed using UAVs, DSLRs, and phone cameras. IDID was a dataset of defective insulators derived from an archival competition containing three classes: flashover damage insulator shell, broken insulator shell, and good insulator shell.

## 2.2. Vision-Based Inspection System

Many studies on inspection systems have been published in the last two decades, including walking PTLIRs [3–5,32–35], flying PTLIRs (UAVs) [6,7], and flying-walking (hybrid) PTLIRs [36–40]. The inspection methods using inspection platforms mainly include vision-based, thermal, radar-based, multi-sensor, etc.

Hydro-Québec [36] proposed a LineDrone technology: a camera and a LIDAR assist the operator for the hybrid PTLIR landing on the power line. Xiang Yue [37] achieved localization and recognition of the autonomous robot by fusing a monocular camera and an encoder. Han Wang [39] designed an inspection robot system with hybrid operation modes to recognize and locate using a camera and a LIDAR. Wenkai Chang [40] designed a hybrid inspection robot loaded with a swingable 2D laser range finder (LRF) to provide an effective recognition basis for robot landing and obstacle crossing. Bo Jia [38] designed a hybrid PTLIR with a walking and flying mechanism, which was more stable than other hybrid PTLIRs since the center of gravity is below the flight plane. The robot has two binocular cameras and an inspection camera for locating and detecting. Hybrid PTLIRs combine their advantages of walking PTLIRs and flying PTLIRs, allowing flying over obstacles and walking along the line for obtaining high-quality data. Visual-based systems are economical and easier to implement in inspection systems. Therefore, hybrid PTLIRs with vision-based systems are a promising and affordable way to inspect PTL system.

## 2.3. YOLO Series

YOLO is a regression method based on deep learning to directly obtain object position and class. YOLO performs prediction on an image and outputs all recognized results at once, without the process of candidate region extraction, so it is fast.

YOLOv1 [22] learned the GoogleNet classification network structure, including 24 convolutional layers and 2 fully connected layers. Although the detection speed of YOLOv1 was fast, the localization accuracy and the recall rate were low. YOLOv2 [23] improved on YOLOv1 using a series of optimization strategies (multi-scale training, a new network, Darknet-19, and added anchor) to increase accuracy while maintaining the original speed. YOLOv3 [24] improved YOLOv2 by using a new network structure, Darknet-53, fusing FPN, and using logistic regression instead of softmax as a classifier. The neck structure of YOLOv4 [25] used the SPP modules, and the Mosaic data augmentation was used for the input part. YOLOv5 [41] added self-adapting anchor frames at the input, using the focus structure and CSP structure at the backbone and the FPN+PAN structure at the neck. The YOLOX [26] was to decouple the head based on the architecture of YOLOv3. The anchor-free and the SimOTA were also proposed. PP-YOLOE [27] improved anchor-free and proposed the novel RepResBlock, ET-head, and dynamic label assignment algorithm TAL. The structure of YOLOv7 [28] was similar to YOLOv5, and the efficient aggregation networks ELAE(E-ELAN), the reparametrized convolutional (RepConv), and the auxiliary head detection were proposed.

YOLOv5 has been updated to v6.1 version; its accuracy, functionality, and robustness have greatly improved over the first version. Algorithms after YOLOv5, such as YOLOX, PP-YOLOE, and YOLOv7, have multiple heads in the head layer, and it is relatively easy to verify the YOLOv5 fusion depth-attention mechanism.

## 2.4. Attention Mechanism

Attention mechanisms can be easily added to neural networks. In the vision field, some popular attention mechanisms are the following: spatial domain [42–44], channel domain [45–48], and mixed domain [49–51]. Each neural network layer could output a feature matrix of size C $\times$ H $\times$ W. The spatial domain attention mechanism adjusts the weights of H and W, the channel domain attention mechanism adjusts the weights of C, and the hybrid domain attention mechanism adjusts the weights of C, H, and W.

Squeeze-and-excitation (SE) [45] mainly learned the importance of each feature channel and the interdependence between channels, which was rather common in the field of

vision. Convolutional block attention module (CBAM) [49] was an extension based on SE, adding a two-dimensional spatial attention matrix. Its pooling method combined global average pooling and global maximum pooling, which can extract more valid information. Coordinate attention (CA) [47] decomposes channel attention into two one-dimensional feature encoding processes that aggregate features along two spatial directions. Remote dependencies can be captured along one spatial direction, while precise location information can be retained along the other spatial direction. Then, the generated feature maps were encoded as a pair of direction-aware and position-sensitive attention maps, respectively. It can be applied complementarily to the input feature map. Efficient channel attention (ECA) [48] was an improved version of SE that uses a $1 \times 1$ convolutional layer after the global average pooling layer, removing the fully connected layer. The module avoids dimensionality reduction and effectively captures cross-channel interactions.

The attention mechanism is a resource allocation mechanism, and the allocated resources are the weights in deep neural networks. The backgrounds and targets of the collected images using FPTLIR have significant depth difference characteristics, and the weights of the feature matrix can be assigned by the differences.

### 3. Hardware and Data

#### 3.1. System Structure

FPTLIR [38] is a developed robot for the inspection of PTL systems. As shown in Figure 1, the FPTLIR mainly consists of flying and walking mechanisms. The structure of the flying mechanism is similar to a hexacopter, mainly consisting of a controlling box, an undercarriage, three batteries, six rotors, and six rotor motors. The batteries are placed under the undercarriage to adjust the center of mass of FPTLIR below the flight plane. The walking mechanism includes a rolling component and a pressing component. The rolling component drives FPTLIR to walk along the line, mainly including the driving and auxiliary wheels. The pressing component provides the pressing force to improve the anti-slip performance of FPTLIR, mainly including the pressing wheels, pressure sensor, cross rod, and linear stepper motor. The FPTLIR can fly up and down the lines and walk along the line with good stability to collect high-quality data, providing a good basis for fitting recognition.

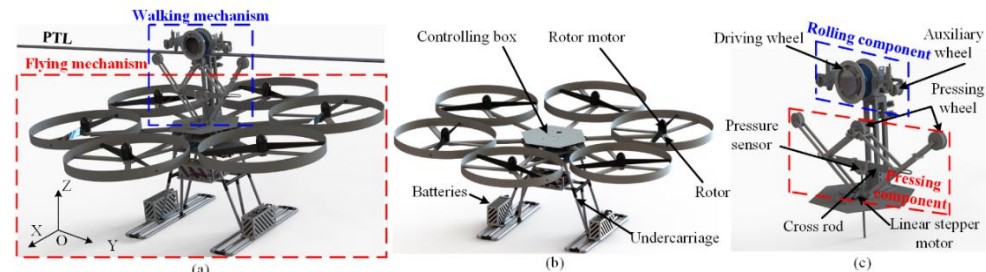

**Figure 1.** Schematic diagram of robot structure. (**a**) complete machine structure; (**b**) flying mechanism; (**c**) walking mechanism.

#### 3.2. System Integration

The load capacity and installation space of FPTLIR are considered in the system integration, and the key components are listed in Table 2. The FPTLIR's size is 1.76 m $\times$ 1.76 m $\times$ 1.1 m, the weight is 38 kg, the rated lift is 42 kg, and the maximum lift is 90 kg. The onboard computer uses NVIDIA jetson NX to run various algorithms and communicates with PX4 and STM32, where PX4 controls the flying mechanism and STM32 controls the walking mechanism. The FPTLIR has two D-RGB cameras and one inspection camera, D435i, ZED, and SJ4000, respectively. Figure 2 shows the installation position of these cameras. The direction of ZED is the same as the X-axis, estimating the position relationship between the transmission line and the machine body when FPTLIR flies up the line. The SJ4000 is installed on the three-axis stabilization platform underneath FPTLIR for inspection targets.

The direction of D435i is the same as the Y-axis for recognizing and locating targets. D435i has a size of 90 mm × 25 mm × 25 mm, a depth field of view angle of 87° × 58°, a depth resolution of 1280 × 720, and binocular infrared depth technology.

**Table 2.** The key components of FPTLIR.

| Component Name | | Mode | Amount |
|---|---|---|---|
| FPTLIR body | | Flying mode | 1 |
| | | Walking mode | 1 |
| Cameras | D-RGB cameras | D435i | 1 |
| | | ZED | 1 |
| | Inspection camera | SJ4000 | 1 |
| Controllers | Central control | NVIDIA Jetson NX | 1 |
| | Walking mode control | STM32 | 1 |
| | Flying mode control | PX4 | 1 |

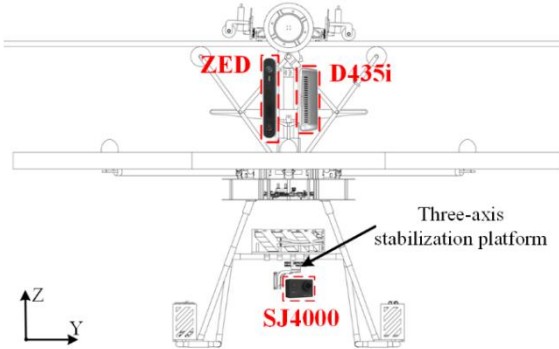

**Figure 2.** The installation position of cameras.

RGB images and depth maps are collected by D435i (there are official modules for the alignment of both types of images), and they are input into a YOLO model trained using a synthetic dataset, where the depth maps are used as input to an attention mechanism to improve the accuracy of detecting the fixtures.

*3.3. Inspection Data*

The components of PTL system inspection mainly include towers, fittings, and power lines. Among them, the fittings are the most difficult task in power inspection: they are the most numerous in type and number of these three categories. Robot inspection is an excellent method for this difficult PTL inspection task.

FPTLIR is a hybrid robot in the mentioned inspection robots [8]. Figure 3 shows the inspection paths of FPTLIR on the PTLs, which can be divided into four steps: flying up-line, walking along the line, crossing obstacles, and flying down-line. The high-quality data is collected when FPTLIR walks along the line.

Figure 4 shows the position relationship between FPTLIR and fittings. O-XYZ, $O_i$-$X_iY_iZ_i$, and $O_r$-$X_rY_rZ_r$ are the global coordinate system, the fitting coordinate system, and the robot coordinate system, respectively. The Z-axis of the global coordinate system is vertically upward, and the Y-axis is perpendicular to the transmission line. The $z_r$-axis of the robot coordinate system is perpendicular to the flying plane, and the $y_r$-axis is the same as the direction of travel in robot walking mode. The initial direction of the fitting coordinate system is the same as the global coordinate system. The coordinate origin of the fitting is at the most edge of the fixture. Since FPTLIR walks along the ground line to collect data, $t_y$ and $t_z$ are related to the power tower structure. $t_x$ is the distance between FPTLIR and the fittings on the X-axis; $\alpha$, $\beta$, $\gamma$ are the rotation angles of the fitting on the $x_i$, $y_i$ and $z_i$ axes, respectively. *l* is the distance between the background and fittings on the

X-axis, and this distance is large, which is the reason for the large difference in depth maps between the fittings and the backgrounds.

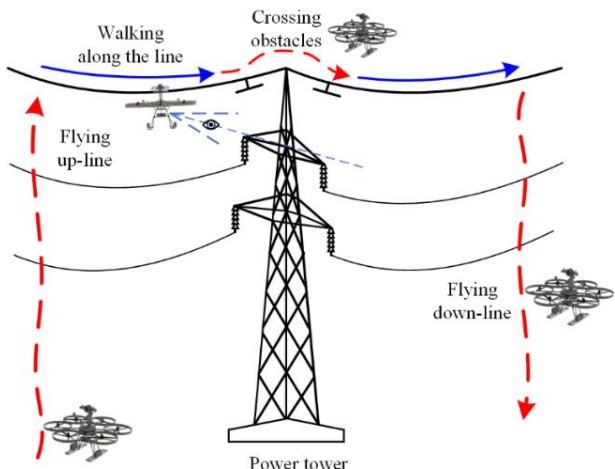

**Figure 3.** Illustration of inspection paths of FPTLIR.

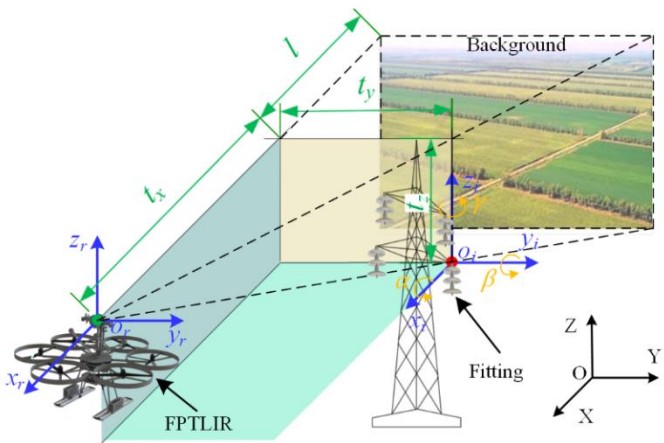

**Figure 4.** The position relationship between FPTLIR and fittings.

Figure 5 shows an RGB image, a depth map of a power tower and some usual fittings. (The infrared heat map is used here to illustrate the depth map because they are very similar.) The difference between the power tower and the background is very clear in the depth map. The depth information of the fittings (e.g., clamps, dampers, insulators) is included in the highlighted portion of the depth map. The large difference in depth information feature could improve the recognition accuracy and reduce the background misdetection when the robot recognizes the fittings.

The inspection data of FPTLIR includes RGB images, depth maps, location information, walking path information, etc. As seen from FPTLIR's motion path in Figure 3, the collected data by FPTLIR is continuous in time and space, but different type data are not continuous. So, this information is matched in time and space, and a complete database of the PTL corridor can be obtained.

The data collection is based on FPTLIR platform, and the high-quality data are collected when FPTLIR walks along the ground line. The collected data has the following characteristics:

- The background of collected images by FPTLIR is mostly ground, and the data is continuously collected along the line to obtain a complete database of PTL corridor.
- The robot's viewpoint is located at the top ground line with the oblique downward view angle, and the collected depth maps have a significant depth difference between the target and background.

When other hybrid PTLIRs are equipped with the same devices along the lines to collect images and depth maps, the collected data has the same above characteristics.

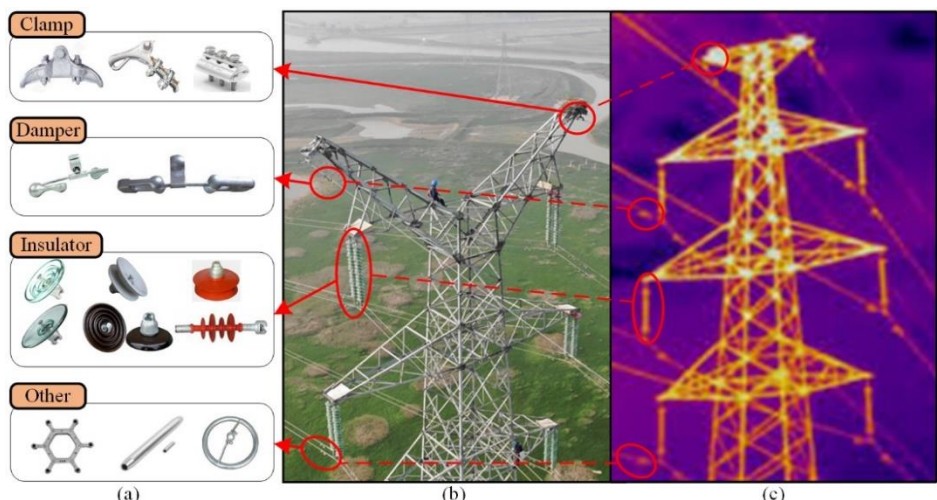

**Figure 5.** Schematic diagram of the collected data. (**a**) some of the fittings; (**b**) RGB image of the power tower; (**c**) depth map of the power tower.

## 4. Proposed Approach

To more accurately recognize the fittings on PTLs, the fitting recognition approach is proposed combining depth-attention YOLOv5 and a prior synthetic dataset. As shown in Figure 6, the proposed approach includes synthesizing the prior synthetic dataset, training the YOLOv5 model, and fusing the depth-attention. First, the prior synthetic datasets are synthesized by formulating synthetic rules from collecting prior series data about the targets. Then, the YOLOv5 model is trained. Finally, the depth-attention mechanism is fused in the detection stages of the YOLOv5 model to recognize fittings.

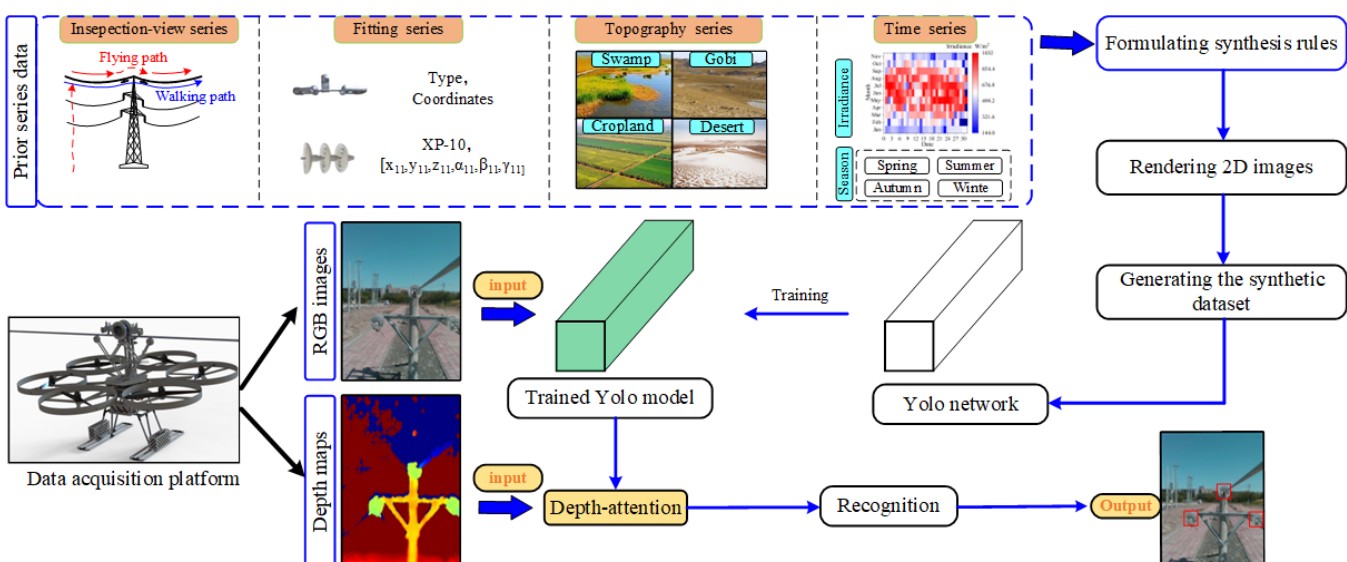

**Figure 6.** Diagram of the fitting recognition approach.

### 4.1. Prior Synthetic Dataset

Prior synthetic datasets based on prior series data are promising for training deep learning models to recognize fittings on PTLs. We have achieved some findings in the previous studies [31,52]. The YOLOX deep learning model is trained using a prior synthetic

dataset including 10,800 images of 720 × 1280 pixels, and the trained model achieves 98.38% of mAP5090 on a test dataset containing 1200 real images.

Figure 7 describes the major synthesis process for a prior synthetic dataset. First, the synthesis rules are formulated based on prior series data. Then, the synthesis rules, insulator models, and HDRI files are input into the Blender (Blender is a programmable modeling software for data synthesis platforms) to auto-synthesize images and annotations. Finally, the images and annotations are collated to generate the prior synthetic dataset.

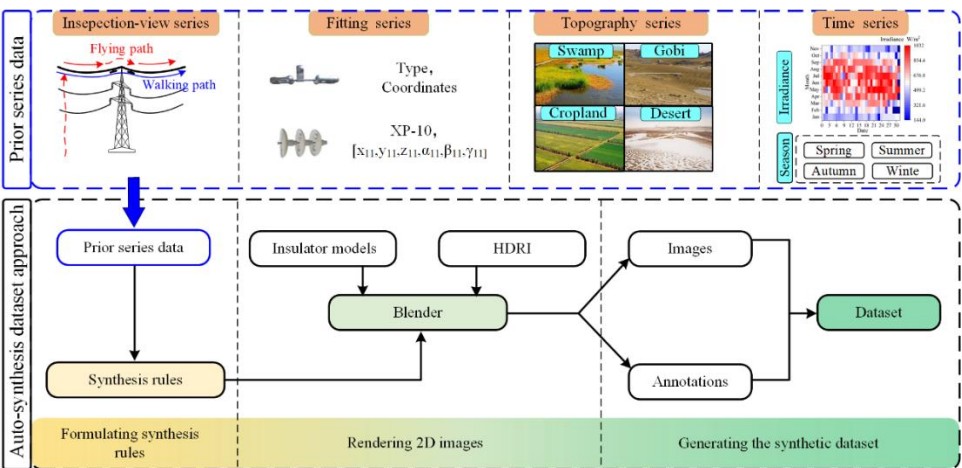

**Figure 7.** The major synthesis process for prior synthetic datasets based on a prior series data.

The synthesis rules are formulated based on prior series data collected on the PTLs by FPTLIR, so the prior synthesized dataset contains more features of the actual inspection images. The trained model could obtain better results using the prior synthesized dataset with less volume. The prior series data include a prior fitting series, a prior inspection-view series, a prior topography series, and a prior time series. The prior fitting series includes the type and location of the fitting; the prior inspection-view series includes the motion path of the robot viewpoint; the prior topographic series includes geographic information about the PTL corridor; the prior time series includes illumination and seasons with time as a variation. The synthesis rules mainly constrain the image background and the viewpoint–target position relationship. The image background is jointly influenced by the prior topography series and the prior time series, and the viewpoint–target position relationship is jointly influenced by the prior fitting series and the prior inspection-view series. Figure 4 illustrates the positional relationship between the viewpoint and the fitting. We fix the three-dimensional coordinates of the view at origin and move the fitting, and their movement rules in Blender are as follows. The four-order translation matrix of the fitting is:

$$T_{xyz} = \begin{bmatrix} 1 & 0 & 0 & t_x \\ 0 & 1 & 0 & t_y \\ 0 & 0 & 1 & t_z \\ 0 & 0 & 0 & 1 \end{bmatrix} \tag{1}$$

where $t_y$ and $t_z$ are related to the power tower structure. $t_x$ is the distance between FPTLIR and the fittings on the X-axis of the global coordinate system.

The four-order rotation matrix of fitting along the xyz axis, $R_x$, $R_y$, $R_z$ are:

$$R_x = \begin{bmatrix} 1 & 0 & 0 & 0 \\ 0 & \cos\alpha & -\sin\alpha & 0 \\ 0 & \sin\alpha & \cos\alpha & 0 \\ 0 & 0 & 0 & 1 \end{bmatrix}; R_y = \begin{bmatrix} \cos\beta & 0 & \sin\beta & 0 \\ 0 & 1 & 0 & 0 \\ -\sin\beta & 0 & \cos\beta & 0 \\ 0 & 0 & 0 & 1 \end{bmatrix}; R_z = \begin{bmatrix} \cos\gamma & -\sin\gamma & 0 & 0 \\ \sin\gamma & \cos\gamma & 0 & 0 \\ 0 & 0 & 1 & 0 \\ 0 & 0 & 0 & 1 \end{bmatrix} \tag{2}$$

where $\alpha$, $\beta$, $\gamma$ are the rotation angles of the fitting on the $x_i$, $y_i$, and $z_i$ axes of the fitting coordinate system, respectively.

To control the fitting within the output image, the rotation angles $\theta_x$ and $\theta_z$ of the viewpoint along the X and Z axes of the global coordinate system are:

$$\begin{cases} \theta_z = \arccos\dfrac{t_z^2 + t_y\sqrt{t_x^2 + t_y^2}}{t_x^2 + t_y^2 + t_z^2} \\ \theta_x = \arctan\dfrac{\sqrt{t_x^2 + t_y^2}}{-t_z}(t_x \geq 0) \\ \theta_x = -\arctan\dfrac{\sqrt{t_x^2 + t_y^2}}{-t_z}(t_x < 0) \end{cases} \tag{3}$$

We could move the fitting in Blender by $R_x$, $R_y$, $R_z$, and $T_{xyz}$ and rotate the view by $\theta_x$ and $\theta_z$.

Algorithm 1 lists the pseudo-code of the main synthesis process. Figure 8 shows an example of the synthesized dataset.

---

**Algorithm 1** Auto-synthesis dataset algorithm

---

for hdris in hdri_file: # Traversing hdri files
   for k in range(1, num_hdri): # Number of cycles per hdri
      hdri_adjust() # Adjustment of environmental parameters
      move() # Mobile fitting
light() # Adjusting ambient light
     save(save_name_image ) # Output RGB image
bpy.data.worlds["World"].node_tree.nodes["Background"].inputs [1].default_value = 1 #
Disconnecting background nodes
     save(save_name_nobackground_image ) # Output background-free image
cv_label() # Automatic annotation

---

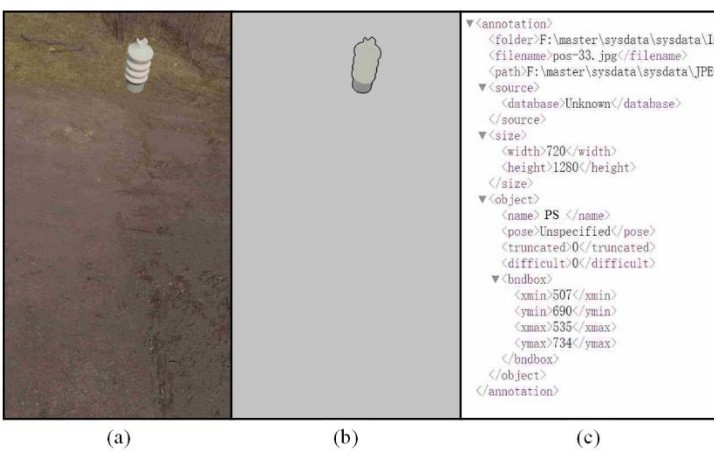

     (a)          (b)          (c)

**Figure 8.** Example of a prior synthetic dataset. (**a**) Normal image; (**b**) backgrounded-free image; (**c**) annotation.

### 4.2. YOLOv5 Network Model

Currently, the latest version of YOLOv5 is v6.1, with mAP5095 at 49% on the COCO dataset. YOLOv5 contains five architectures, ordered by size from smallest to largest as YOLOv5n, YOLOv5s, YOLOv5m, YOLOv5l, and YOLOv5x. The YOLOv5s architecture is adopted considering accuracy, real-time, and lightweight of models deployed on FPTLIR. Figure 9 shows the main structure of the YOLOv5 model, including input, backbone, neck, and head.

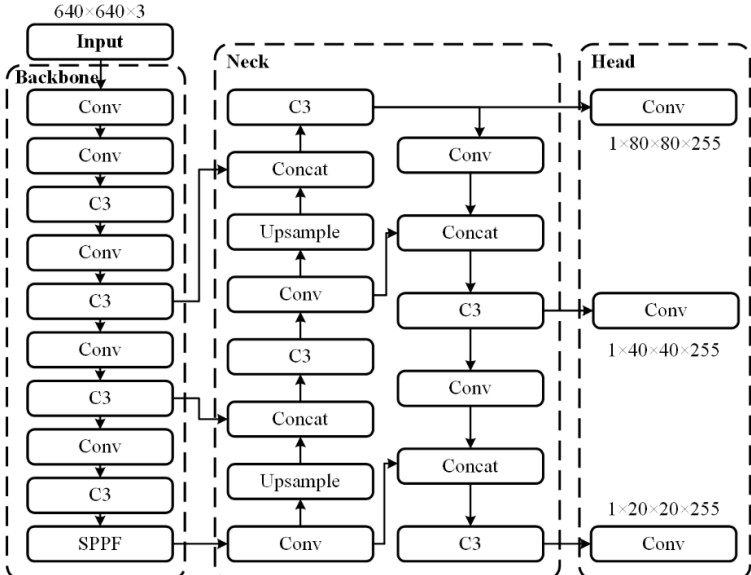

**Figure 9.** YOLOv5 network structure.

The input includes self-adaptive image scaling, Mosaic data enhancement, and self-adaptive anchor box calculation. Different images are scaled to fixed size by self-adaptive image scaling with normalization. The Mosaic data enhancement uses four images and stitches them together in a randomly scaled, cropped, and matched manner. This enhancement method can combine several images into one, enriching the dataset and improving the training speed of the model. Meanwhile, it could reduce the memory requirement of the model.

The backbone benchmark network is usually some high-performance classifier to extract general features. The CSPDarknet53 structure is introduced into the backbone, and the focus structure is used as the backbone to crop the input image by slice operation and stitch in the channel dimension, reducing the params and FLOPs. The focus is replaced with $6 \times 6$ sized convolutional layers in YOLOv5 v6.0, which are equivalent. However, the convolutional layers are more efficient for GPU devices.

The neck is located in the middle of the backbone and head to further improve the features' diversity and output the enhanced effective feature.

Head is the classifier and regressor of YOLOv5 for predicting the location and classification of targets. The enhanced effective feature layers obtained by the backbone and neck are processed by $1 \times 1$ convolution in the head to output three feature layers. As shown in Figure 10, the size of the feature layer matrix is $S \times S \times (B \times (5 + C))$. The feature matrix is divided into $S \times S$ grid cells; each grid cell has B anchor boxes; each box includes center coordinates (x and y), width (w), height (h), confidence, and probability of C classes. The prediction box is obtained by decoding the anchor box. Equation (4) shows the total loss $L_{\text{total}}$ of the interaction between the prediction box output results and the real box.

$$L_{\text{total}} = \sum_i^N (\lambda_1 L_{\text{box}} + \lambda_2 L_{\text{obj}} + \lambda_3 L_{\text{cls}}) = \sum_i^N (\lambda_1 \sum_j^{B_i} L_{CIoU_j} + \lambda_2 \sum_j^{S_i \times S_i} L_{obj_j} + \lambda_3 \sum_j^{B_i} L_{cls_j}) \quad (4)$$

where $N$ is the number of detection layers, $B_i$ is the number of annotations assigned to the anchor box, $S \times S$ is the size of grid cells. $L_{\text{box}}$, $L_{\text{obj}}$, and $L_{\text{cls}}$ are the bounding box regression loss, the target object loss, and the classification loss, respectively. $\lambda_1$, $\lambda_2$, and $\lambda_3$ are the weights of these three losses, respectively.

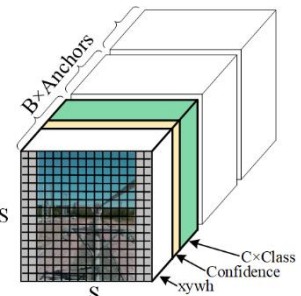

**Figure 10.** Output feature matrix of YOLOv5 head.

### 4.3. Depth-Attention Mechanism

The attention mechanism is realized by adaptively weighting features according to the importance of the input in a visual area, such as SE, CBAM, ECA, CA, Non-Local, and GCNet. For YOLOv5 networks, the attention mechanism is usually introduced at the last layer of the backbone or in the C3 module. This process [53] is obtained:

$$Attention = f(g(x), x), \tag{5}$$

where $x$ is an input feature, $g(x)$ is input signal weighting, and $f(g(x),x)$ represents the process of processing the input feature $x$ based on the weighting $g(x)$.

#### 4.3.1. Principle of Depth-Attention

Figure 11 illustrates the network structure of the depth-attention YOLOv5. The depth-attention mechanism is introduced after the head and frozen during training, differing from other attention mechanisms described above. It can also be directly introduced after the trained model.

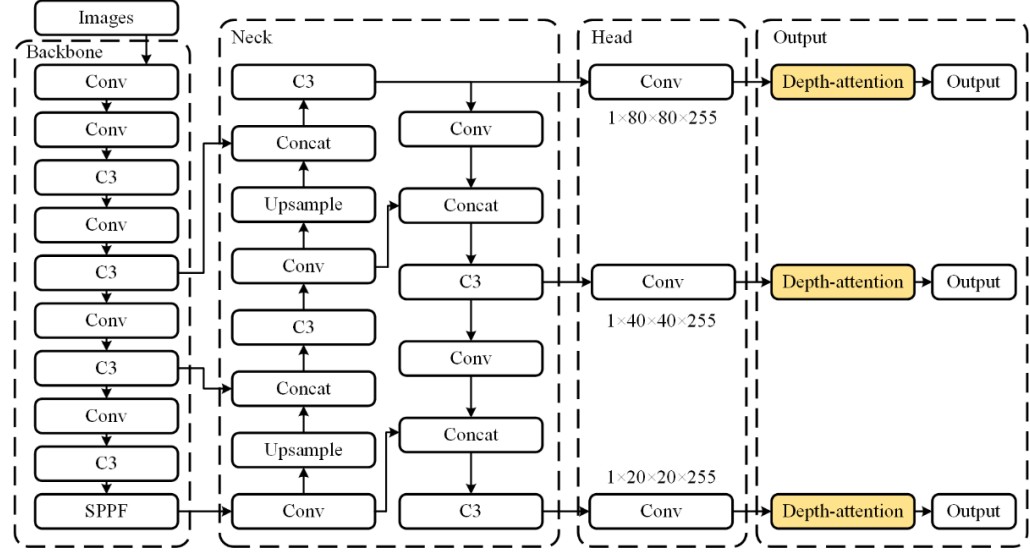

**Figure 11.** The structure of depth-attention YOLOv5. The network consists of four main parts: input, backbone, neck, head, and output. (The yellow part is the added depth-attention).

Figure 12 and Equation (6) show the detailed principle of the depth-attention in YOLOv5. $f(g(x),x)$ denotes that the confidence and classification scores in $x$ are processed by $g(x)$. It is defined as:

$$f(g(x), x) = x[:,:,i,j,4:] + g(x)[i,j,1] \tag{6}$$

where $x$ is the result of head output with five dimensions. The first dimension is the batch size, which is the number of input images simultaneously. The second dimension is the number of anchor boxes. The third and fourth dimensions are grid cell coordinates ($i, j$). The last dimension includes the center point coordinates, the prediction box's width and height, confidence, and the scores of the four classifications of targets, respectively. $g(x)$ is a depth-attention matrix of size S × S obtained by processing the depth map, representing the distribution of attention on the image.

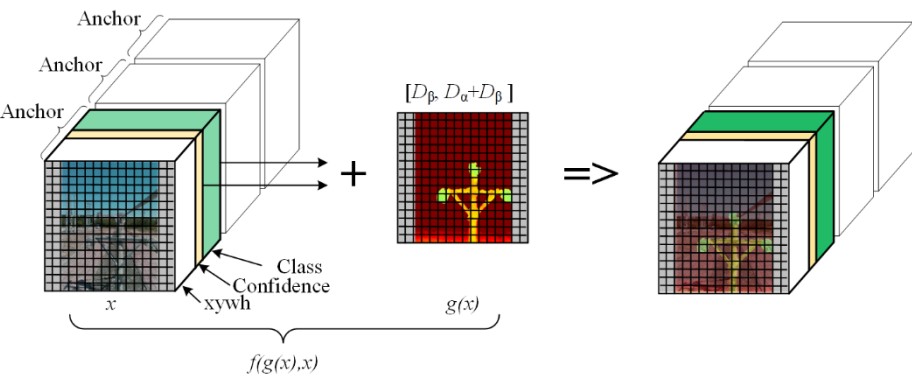

**Figure 12.** Principle of depth-attention mechanism.

### 4.3.2. Depth-Attention Matrix

The images and the depth maps are collected by the D-RGB camera D435i. The RGB images and depth maps need to be cropped and matched, making them the same time and size. RGB images are used as model inputs, and depth maps are used as attention inputs. The depth-attention matrix is:

$$g(x) = \text{sigmoid}(\text{avgpool}(Dmap) + \text{maxpool}(Dmap)) \times D_\alpha + D_\beta \tag{7}$$

The depth map is saved as a JET pseudo-color image, and *Dmap* is a two-order matrix obtained by inverse pseudo-color processing of the depth map, with the range of 0~255. The width and height of the pooled *Dmap* matrix in Equation (7) are consistent with the size of $x$ sigmoid normalizes the pooled *Dmap* matrix. $D_\alpha$ is the attention coefficient, adjusting the importance of attention on the results. $D_\beta$ is an offset coefficient, adjusting the attentional share of the background and the target, and its absolute value does not exceed $D_\alpha$. The range of $D_\alpha$ and $D_\beta$ and taking values is flexible due to the depth difference between background and target.

## 5. Experiments and Results

To verify the feasibility and validity of the proposed approach, we conducted the test field experiments and line field experiments, respectively.

### 5.1. Test Field Experiment

To verify the feasibility of depth-attention in YOLOv5, we synthesize a fitting dataset and train a depth-attention YOLOv5 model. The fitting dataset includes insulators (XP-10), suspension clamp (XG-4028), strain clamp (NLD-4), and damper (FR-10). A test dataset is collected around the fittings at 3 m to 5 m using the D435i camera. The test dataset includes only the suspension clamp, but the recognition classes of the trained model include four classes of fitting dataset to increase the probability of misdetections and omissions, highlighting the role of depth-attention.

The tests are summarized as follows:

- A fitting dataset is synthesized, and a standard YOLOv5 model is trained.
- A depth-attention is introduced in YOLOv5, comparing the recognition results of the standard YOLOv5 and depth-attention YOLOv5 on the test.

### 5.1.1. Datasets

The synthetic dataset is divided into training and validation datasets, with a ratio of 9:1. The total volume of the dataset is 4000, and the volume of each class is 1000. The volume of the test dataset is 200, including real RGB images and corresponding depth maps with a size of 640 × 480. Figure 13 shows a partial example of all datasets.

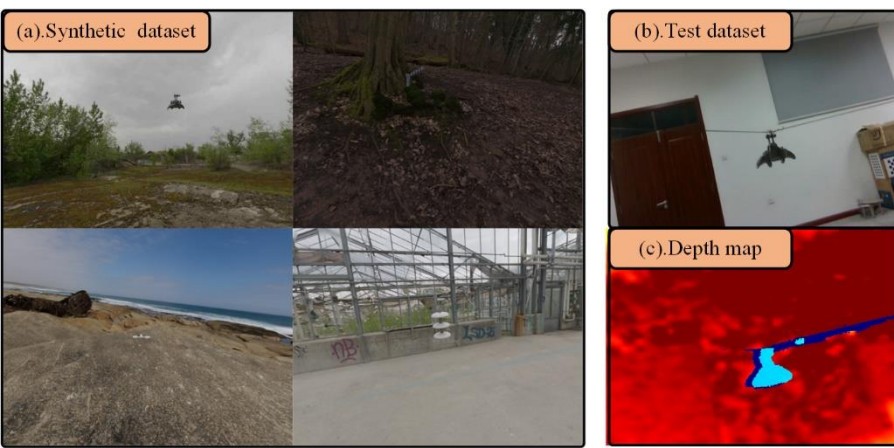

**Figure 13.** Example of the dataset.

### 5.1.2. Training

The initial learning rate of the YOLOv5 network is 0.01; the weight decay coefficient is 0.0005; the warmup momentum is 0.80; the epoch number is 300. The total loss and mAP5095 during the training process are shown in Figure 14. The loss and mAP tend towards stability after 200 epochs, indicating that the model was successfully trained.

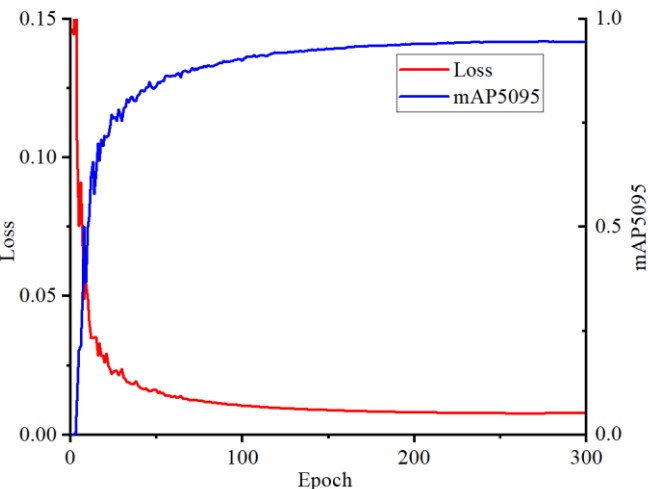

**Figure 14.** Loss and mAP during training of YOLOv5 model in the test field experiments.

### 5.1.3. Comparison between YOLOv5s and Our Proposed Method

The mAP is used as a metric for recognition accuracy, which is the average of average precisions (APs) with IoU thresholds greater than a setting threshold. COCO's mAP recognition standard is mAP5095 (average mAP on step size 0.05 for thresholds from 0.5 to 0.95). The $AP$ is calculated by precision ($P$) and recall ($R$) as:

$$AP = \int_0^1 P(R)\mathrm{d}R \tag{8}$$

The *P* and *R* are:

$$P = \frac{TP}{FP + TP} \tag{9}$$

$$R = \frac{TP}{FN + TP} \tag{10}$$

True positive *TP* means the content in the bounding box predicted by the model is the insulators, and the result is correct. False positive *FP* means the content in the bounding box predicted by the model is the insulators, and the result is false. True negative *TN* means the content in the bounding box predicted by the model is not the insulators, and the result is correct. False negative *FN* means the insulator is not recognized.

Table 3 compares the results of the standard YOLOv5 model and the depth-attention YOLOv5 model with $D_\alpha = 3$ and $D_\beta = -6$ on the test dataset. It can be seen that depth-attention improves the mAP5095 of the YOLOv5 model by 5.2%, and inference speed is only 3.1 s slower, verifying the feasibility of the depth-attention.

**Table 3.** Comparison of the YOLOv5 and depth-attention YOLOv5.

| Model | Fitting | mAP 50(%) | mAP 5095(%) | P | R | Inference Speed (ms) |
|---|---|---|---|---|---|---|
| YOLOv5 | XG | 96% | 62.9% | 0.942 | 0.935 | 7.9 |
| Depth-attention YOLOv5 | XG | 99% | 68.1% | 0.983 | 0.983 | 11 |

To further elaborate the process of depth-attention to improve accuracy, we analyzed the last layer's feature maps. Figure 15a is the input image of the test dataset. Figure 15b–d visualize the depth-attention matrix, $g(x)$, with the same size as the feature maps of the last layer, and the brightness is the value of $g(x)$. Figure 15e,f show the feature heat map of the last layer at the standard YOLOv5 inference, including both correct and false inferences. Figure 15g,h are the feature heat map at the depth-attentionYOLOv5 inference. Comparing Figure 15f,g, the weight of the misdetection is reduced by depth-attention. Figure 16 shows more understandable inference results: YOLOv5 falsely recognized the class NLD, and depth-attention reduced the confidence of false recognition and improved the confidence of correct recognition of XG. The results indicate that the depth-attention mechanism could improve the accuracy of recognizing fittings and reduce the probability of misdetection.

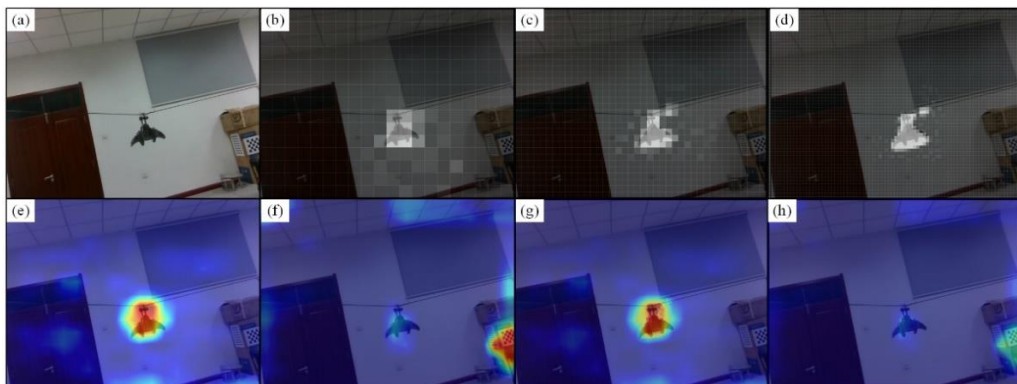

**Figure 15.** Visualization of feature maps and depth-attention matrices. (**a**) Input image of the test dataset. Depth-attention matrix, $g(x)$, on the feature map of the last layer: (**b**) $g(x)$ of large target recognition, (**c**) $g(x)$ of medium target recognition, (**d**) $g(x)$ of small target recognition. Feature heat map of the last layer at the standard YOLOv5 inference: (**e**) correct inference, (**f**) false inference. Feature heat map of the last layer at the depth-attention YOLOv5 inference: (**g**) correct inference, (**h**) false inference.

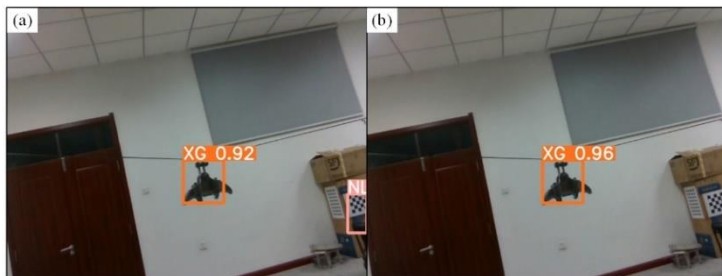

**Figure 16.** Inference results. (**a**) Standard YOLOv5; (**b**) depth-attention YOLOv5.

*5.2. Line Field Experiment*

To test the actual operation of FPTLIR on the line, we use FPTLIR to collect data on an uncharged PTL at 10 KV and compare the recognition results of different attention mechanisms.

### 5.2.1. Test Site

The test site locates in the Shihezi City of Xinjiang, and the test line is a 10 KV PTL consisting of two tension towers at both ends and two tangent towers at the middle, marked as ① and ② in Figure 17, respectively. The towers are 3 m high and 10 m apart. The insulator models of the tension tower are XP-10 and FXBW-10. The insulator models for tangent towers are PS-15 and FPQ-10. The FPTLIR walks along the power line and continuously collects RGB images and depth maps.

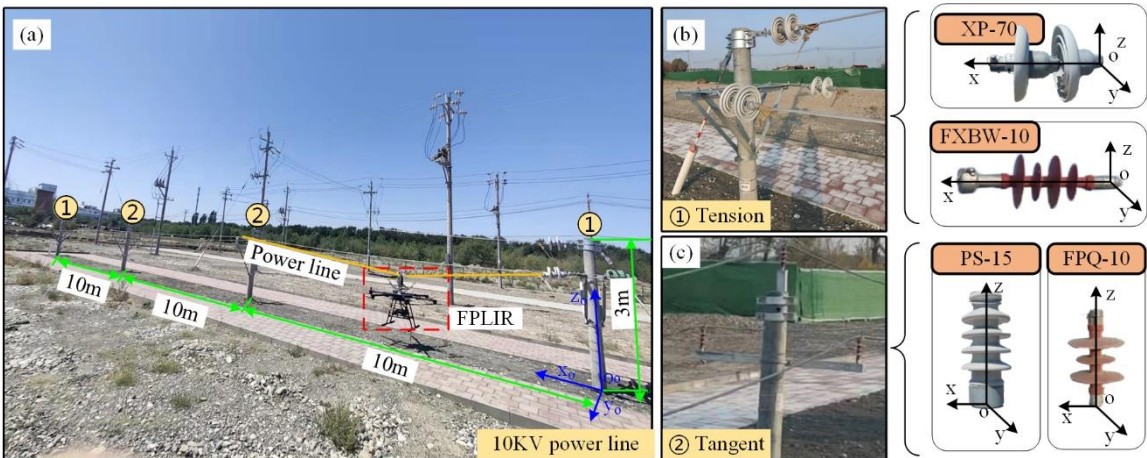

**Figure 17.** Schematic diagram of the line field experiment. (**a**) Overall diagram of the line; (**b**) tension tower; (**c**) tangent tower.

### 5.2.2. Synthetic Dataset

The main prior series data for synthesizing the dataset are as follows:

Viewpoint coordinates: the x-axis and y-axis coordinates are $O_o$, and the z-axis coordinates are randomly fixed at 2 m to 3 m in the global coordinate system $O_o\text{-}x_oy_oz_o$.

Fitting coordinates: the x-axis coordinates change from 1 m to 6 m, the y-axis coordinates change from $-0.5$ m to 0.5 m, and the z-axis coordinates change from 2 to 3 m in the global coordinate system $O_o\text{-}x_oy_oz_o$.

Fitting types include ceramic tension insulator (XP-70), composite tension insulator (FXBW-10), ceramic post insulator (PS-15), and composite post insulator (FPQ-10).

Topography type: The topography is all-terrain without restrictions.

Time data: No restrictions for all seasons and illumination at 0 W/m$^2$ to 1000 W/m$^2$.

The number of generated images for each class of fitting is 1000.

The important parameters are listed in Table 4:

**Table 4.** Important parameters for the prior series data.

| Object | Fitting Coordinates(m) | Fitting Rotation Angle (°) | Viewpoint Coordinates (m) | Illumination (W/m²) |
|---|---|---|---|---|
| XP-70 | x: [1, 6], y: [−0.5, 0.5], z: [2, 3] | $\alpha$: 0, $\beta$: [−30, 30], $\gamma$: [70, 110], [−110, −70] | [0, 0, 2~3] | [0, 1000] |
| FXBW-10 | x: [1, 6], y: [−0.5, 0.5], z: [2, 3] | $\alpha$: 0, $\beta$: [−30, 30], $\gamma$: [70, 110], [−110, −70] | [0, 0, 2~3] | [0, 1000] |
| PS-15 | x: [1, 6], y: [−0.5, 0.5], z: [2, 3] | $\alpha$: [−30, 30], $\beta$: [−30, 30], $\gamma$: 0 | [0, 0, 2~3] | [0, 1000] |
| FPQ-10 | x: [1, 6], y: [−0.5, 0.5], z: [2, 3] | $\alpha$: [−30, 30], $\beta$: [−30, 30], $\gamma$: 0 | [0, 0, 2~3] | [0, 1000] |

Figure 18 shows the synthetic dataset for the four classes of fittings. Figure 19 shows the annotated information of the synthetic dataset, including the coordinates of the center point and the size of the annotation box. The fitting positions of the synthetic dataset are mostly concentrated in the upper part of the image, and the fitting sizes are smaller than 0.2 times the image, mostly concentrating in 0.03 times the image size.

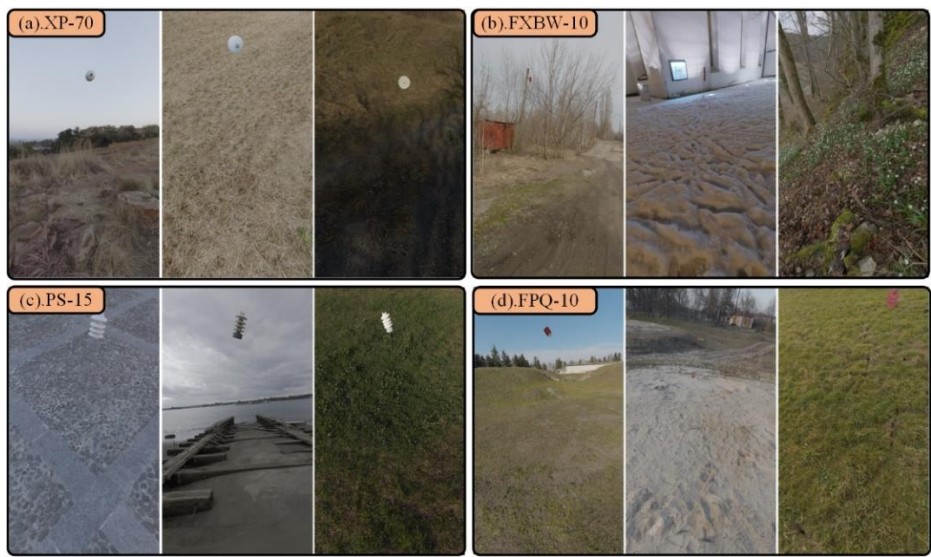

**Figure 18.** Example of the synthetic dataset. (**a**) Synthesis image of XP-10; (**b**) synthesis image of FXBW-10; (**c**) synthesis image of PS-15; (**d**) synthesis image of FPQ-10.

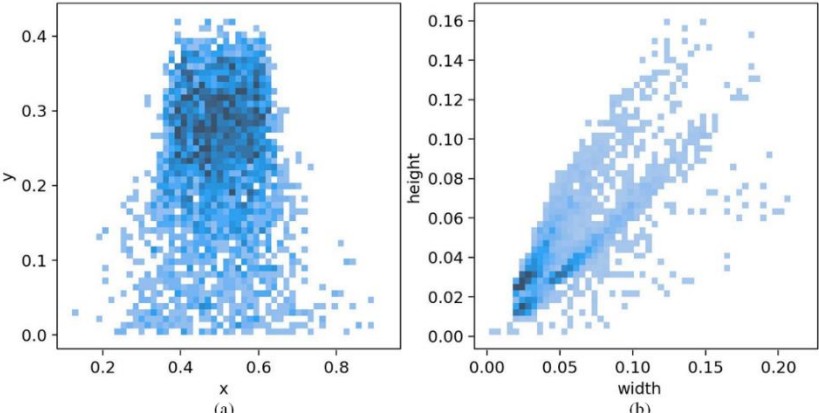

**Figure 19.** The annotated information of the synthetic dataset. (**a**) The coordinates of the center point of the annotation box; (**b**) the size of the annotation box.

The synthetic dataset has a volume of 4000, containing images and annotation files in VOC format. Before training the model, the synthetic dataset is divided into a training set and a validation set with a ratio of 9:1; the annotation format is converted to YOLO.

### 5.2.3. Test Dataset

Figure 20 shows the collected data in the test field, including RGB images and depth maps at $640 \times 480$. The depth maps are displayed using pseudo-color (JET), and the blue part without data is filled with red. The test dataset contains four classes, namely, XP-70 for ceramic tension insulators, FXBW-10 for composite tension insulators, PS-15 for ceramic post insulators, and FPQ-10 for composite post insulators. The total volume of the test dataset is 400, and the volume of each class is 100.

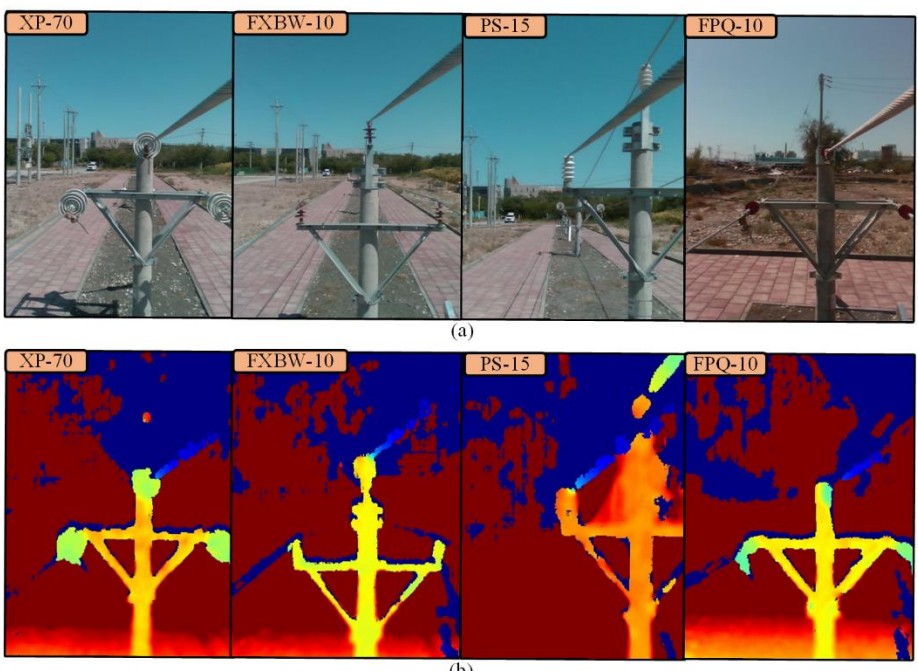

**Figure 20.** Illustration of the test dataset. (**a**) RGB images; (**b**) depth maps.

The test dataset is annotated using labelImg software, saving as YOLO format. The names of RGB images, annotation files, and depth maps are strictly matched.

### 5.2.4. Training

The training and testing run in ubuntu 18.04 LTS system, using a computer with graphics NVIDIA GTX-1660. The dataset is synthesized based on a prior data with volume of 4000 and four kinds. The dataset is given with space–time properties along the PTLs, enabling YOLOv5 model to learn effective features more easily.

YOLOv5 extracts feature maps from the input samples in the backbone stage. Then, the feature maps are enhanced in the neck to obtain a better representation, mapped from feature space to annotation space by the recognition head. The predicted annotation vector has seven channels, including position information (four channels), confidence (one channel), and class (four channels). Three loss values are calculated between the predicted annotation vectors and the true annotations to evaluate the performance of the model parameters in the predicted position, confidence, and class information of the fitting. Finally, the parameters of the model are updated using the stochastic gradient descent (SGD) algorithm.

The training hyperparameters are set as follows: epoch, batch size and initial learning rate are 300, $-1$, and 0.01, respectively.

Figure 21 shows a loss and mAP5095 during the training process. The total loss shows that the loss rapidly decreases in the first 50 epochs and stabilizes at 200 epochs, indicating the trained model using the synthetic dataset with great results and no overfitting.

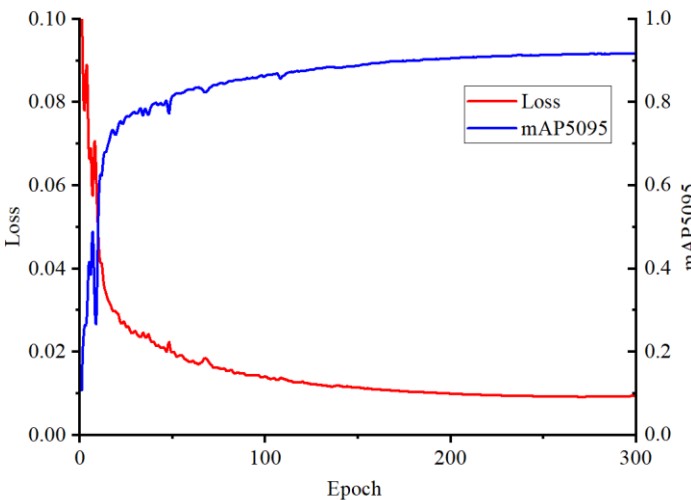

**Figure 21.** Loss and mAP5095 during the training process of the YOLOv5 model in the line field experiments.

### 5.2.5. Experimental Results in Test Dataset

The trained standard YOLOv5 and the depth-attention YOLOv5 model are tested for the test dataset; the results are shown in Figure 22. The standard YOLOv5 achieves mAP5095 at 61.4% on the test dataset, and the depth-attention YOLOv5 achieves mAP5095 at 64.6%, improving by 3.2%. A more intuitive representation is shown for the confusion matrix in Figure 23. The closer the value of the diagonal line is to 1, indicating the fewer omissions. The closer the value of the rightmost and bottom row is to 0, indicating the fewer misdetections. The misdetections of confusing the background as the fitting are significantly reduced; the misdetections of confusing the fitting as the background are also slightly reduced; the omissions of each class are improved.

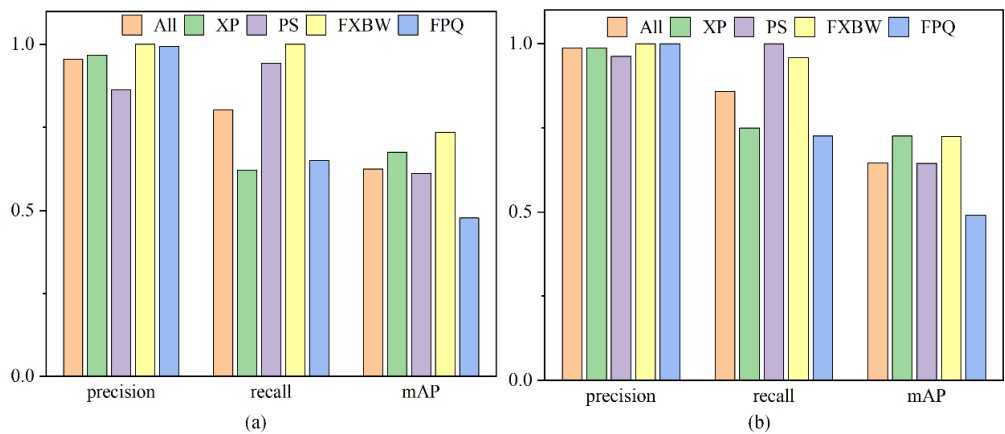

**Figure 22.** Comparison of recognition accuracy. (**a**) YOLOv5; (**b**) depth-attention YOLOv5.

### 5.2.6. Comparison with the Recognition Results Using Different Attentional Mechanisms

We introduce the ECA, CA, and SE attention mechanisms into the last layer and C3 module of the YOLOv5 backbone, respectively, where the better test results with the C3 module in two ways are shown in Figure 24 and Table 5. All results below 10% of the standard YOLOv5 are not displayed in Figure 24 and Table 5. $D_\alpha$ is 2, and $D_\beta$ is $-1$ in depth-attention.

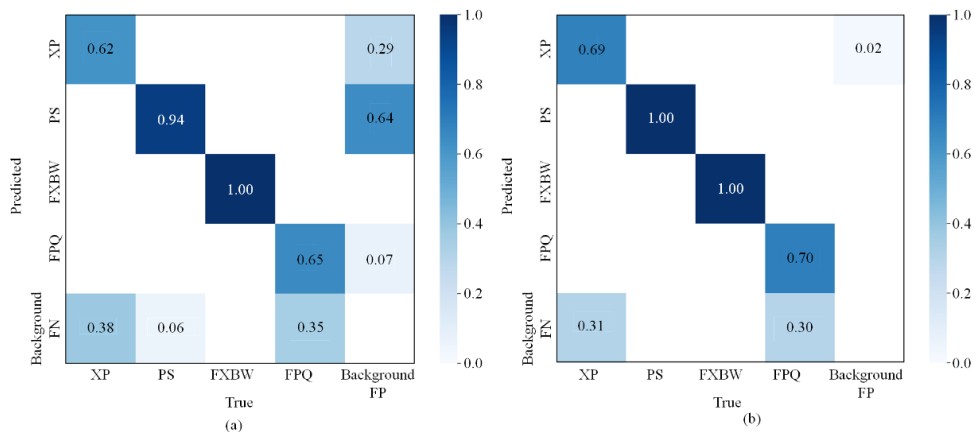

**Figure 23.** Confusion matrix. (**a**) YOLOv5; (**b**) depth-attention YOLOv5.

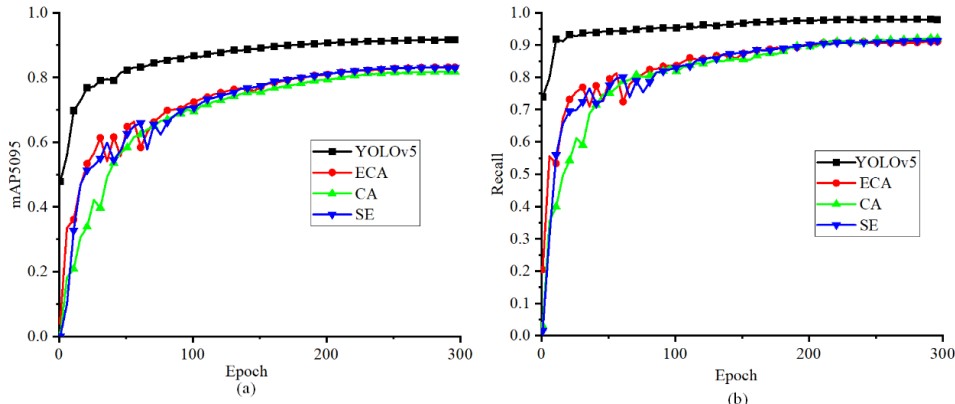

**Figure 24.** Variation trend of training metrics in the experiment. (**a**) mAP5095; (**b**) recall.

**Table 5.** Performance comparison with the YOLOv5 model and the depth-attention YOLOv5 model.

| Model | Add Location | Best Result Category | P | R | mAP 5095(%) | Inference Speed (ms) |
|---|---|---|---|---|---|---|
| YOLOv5 | - | All | 0.863 | 0.87 | 61.4% | 9.1 |
| | | XP | 0.92 | 0.78 | 70.4% | |
| | | PS | 0.59 | 0.99 | 57.6% | |
| | | FXBW | 1 | 0.907 | 69% | |
| | | FPQ | 0.941 | 0.804 | 48.6% | |
| ECA | C3 | PS | 0.943 | 1 | 68.2% | 27.3 |
| | | FXBW | 1 | 1 | 59.4% | |
| CA | C3 | PS | 0.82 | 1 | 67% | 22.4 |
| | | FXBW | 1 | 0.863 | 62.6% | |
| SE | C3 | PS | 0.98 | 1 | 69.2% * | 27.1 |
| | | FXBW | 1 | 1 | 59.7% | |
| Depth-attention YOLOv5 | Head last | All | 0.987 | 0.858 | 64.6% * | 12.6 |
| | | XP | 0.987 | 0.749 | 72.6% * | |
| | | PS | 0.962 | 1 | 64.4% | |
| | | FXBW | 1 | 0.958 | 72.4% * | |
| | | FPQ | 1 | 0.726 | 49.0% * | |

* Best results for improvement.

The mAP5095 of ECA, CA, and SE attention mechanisms have no overall improvement in Figure 24, but the mAP5095 of a single class is improved. Our proposed approach could improve the recognition validity for all classes. Except for the PS class, the improvement

validity is better than other attention mechanisms. The inference speed is 3 ms slower than the standard YOLOv5 model and 10 ms to 15 ms faster than other attention mechanisms.

## 6. Discussion

### 6.1. Time-Cost Synthetic Dataset

The synthetic dataset of the test field experiment contains 4000 images of size $640 \times 480$, and each group data takes an average of 4.8 s (including the time for generating a RGB image, a background-free image, and an annotation), and the complete dataset takes 5.3 h to generate. The synthetic dataset of the line field experiment contains a total of 4000 images of size $720 \times 1280$, and each group data takes an average of 7.7 s, and the complete dataset takes 8.5 h to generate. A more detailed time-cost is shown in Figure 25. It can be seen that the time-cost of the synthetic dataset is very small compared to the cost of collecting the fitting dataset on the actual PTLs.

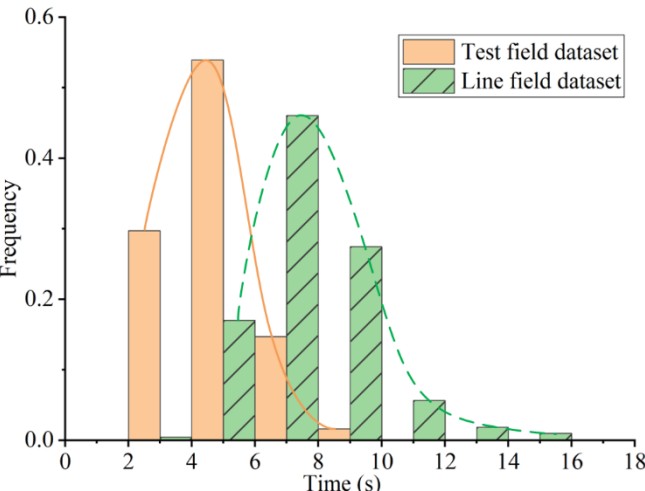

**Figure 25.** The time-costs of each group of synthetic data.

There is a different cost between the two datasets because the time-cost of synthetic datasets is directly proportional to the device's performance and the size of output images, and a superior device can further reduce the cost. Figure 25 shows the same trend for both types of datasets since the time-cost is related to the file size of HDRI. Hardware performance affects the left–right movement of the data in Figure 25, and HDRI file size affects the trend direction of the data.

### 6.2. Influence of Increasing Dataset Volume on Results

Both experiments synthesize the dataset volumes are 4000; on the one hand, the time-cost is reduced; on the other hand, the dataset volume is limited to control the model probability of omission and misdetection, verifying the validity of the depth-attention mechanism more easily.

We increased the dataset volume, generating a synthetic dataset with 8000 volumes of tension insulator XP. The YOLOv5 model is trained again and tested on the line field test dataset, and the comparison results with depth-attention YOLOv5 are shown in Table 6. The results show that increasing the training dataset volume can improve the recognition accuracy of the model, and the depth-attention could still improve the mAP5095 of the standard YOLOv5 with 0.3%.

**Table 6.** Test results on the line test dataset of the YOLOv5 and the depth-attention YOLOv5.

| Model | Fitting | mAP 50(%) | mAP 5095(%) | P | R |
|---|---|---|---|---|---|
| YOLOv5 | XP | 97.8% | 80.5% | 1 | 0.957 |
| Depth-Attention YOLOv5 | XP | 98.8% | 80.8% | 1 | 0.977 |

We checked all recognized images and found that both models have no misdetection. Figure 26 shows all recognition results of YOLOv5 and depth-attention YOLOv5. The number of omissions of YOLOv5 is 14, and the depth-attention mechanism corrects four omissions (orange outer frame marker). It can be found that the omission of the fittings has the same characteristics: the fittings are incomplete. The omission could be inferred because the training dataset includes a few incomplete fittings. As reflected in Figure 19a, the central coordinates of the annotation box at the edge are less. The omission of the fittings score is too small, and the depth-attention mechanism is limited to correct all omissions. It is shown that the magnitude of accuracy enhancement by the depth-attention mechanism is related to a good model base and training data.

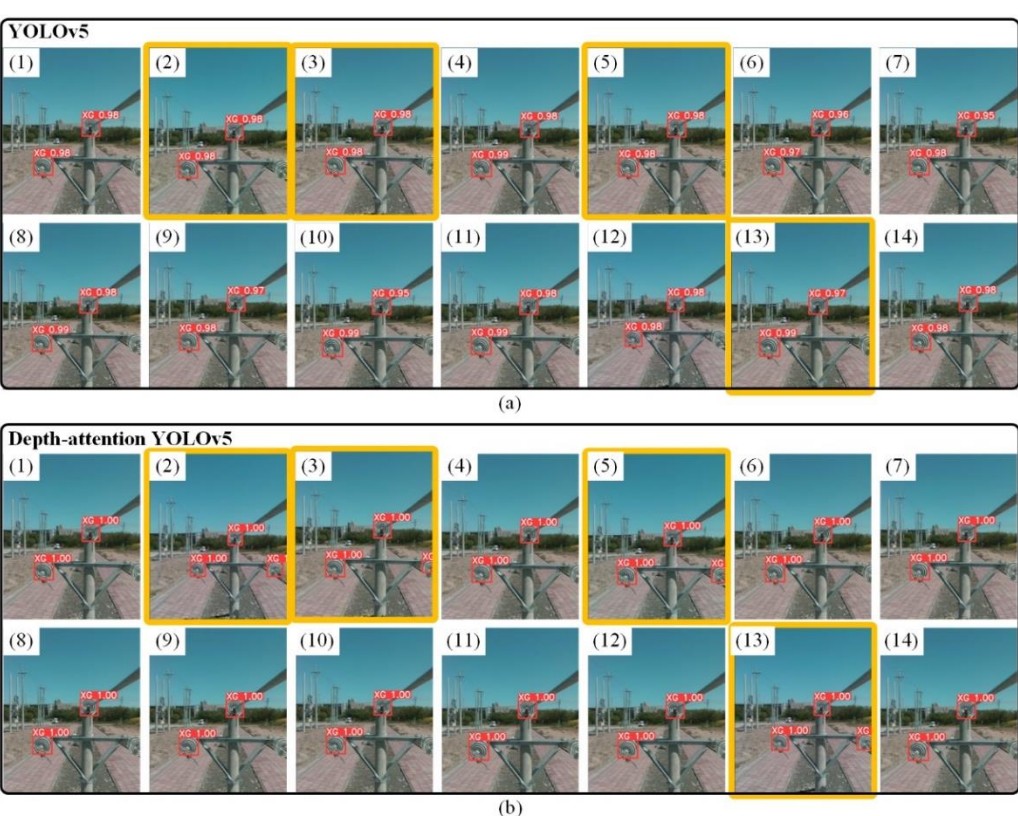

**Figure 26.** Recognition results on the line test dataset. (**a**) YOLOv5; (**b**) depth-attention YOLOv5.

### 6.3. The Effect of $D_\alpha$ and $D_\beta$ on the Depth-Attention Mechanism

For the line experiments, Table 7 shows the mAP5090 on the line test dataset for different $D_\alpha$ and $D_\beta$. The mAP5095 of the standard YOLOv5 model is 61.4%, and different values of $D_\alpha$ and $D_\beta$ have different effects on the depth-attention YOLOv5, with a maximum difference of 1.7% for mAP5095. The effect of depth-attention YOLOv5 is best when $D_\alpha$ is slightly larger than the absolute value of $D_\beta$, interpreted as slightly greater attention to the enhanced target than the diminished background.

### 6.4. Substitution of Depth Information

The principle of the attention mechanism is adaptively weighting features according to the importance of the input. The depth-attention mechanism gives a prior feature matrix to adjust the weights of the background and target. Depth-attention is quite suited for the recognition of a PTL system because of the large depth differences between the target and background in the collected data. The infrared information is similar to the depth map, with significant depth differences between the background and the target. It has the same effect as the depth-attention mechanism, which could replace depth maps of depth-attention mechanism.

**Table 7.** The mAP5090 of depth-attentionYOLOv5 at different $D_\alpha$ and $D_\beta$.

| $D_\beta$ \\ $D_\alpha$ | 1 | 2 | 3 | 4 |
|---|---|---|---|---|
| 0 | 64.30% | 64.60% | 64% | 64% |
| −1 | 64.00% | 64.60% | 64.70% | 63.90% |
| −2 | 63.40% | 64.20% | 64.60% | 64.10% |
| −3 | - | 63.70% | 64.50% | 64.60% |
| −4 | - | 63.10% | 64.20% | 64.60% |
| −5 | - | - | 63.50% | 64.50% |
| −6 | - | - | 63.20% | 64.10% |
| −7 | - | - | - | 63.60% |
| −8 | - | - | - | 63.00% |

The darker the color, the larger the value.

## 7. Conclusions

We propose a fitting recognition approach combining depth-attention YOLOv5 and prior synthetic dataset. The conclusions are as follows.

1. Datasets are automatically synthesized based on prior series data, expediting the efficiency of algorithm development. Three synthetic datasets are generated, including a dataset of 4000 images with a size of 640 × 480 on the test field, a dataset of 4000 images with a size of 720 × 1280 on line field, and a dataset of 8000 XP images with a size of 640 × 480. The synthetic datasets spend 26.4 h, and the synthetic dataset with the largest volume takes 12.6 h.

2. Depth-attention mechanism is proposed based on the depth difference of collected data, applying to target recognition with depth difference information. The test field experiments verify the feasibility of the depth-attention mechanism to improve the accuracy of the YOLOv5 model. Among them, AP, recall, and precision are 68.1%, 98.3%, and 98.3%, increasing by 5.2%, 4.8%, and 4.1%, respectively.

3. The depth-attention YOLOv5 model is trained using a synthetic dataset for real line experiments, and the results show that the depth-attention mechanism acquires a map of 64.6%, improving mAP by 3.2%. Compared with other attention mechanisms, the depth-attention mechanism improves the mAP in all aspects and reduces omission and misdetection, providing better results and faster inference.

The limitations of the proposed approach mainly focus on two aspects: (1) $D_\alpha$ and $D_\beta$ are related to the recognition environment, requiring testing to determine the optimal parameters. (2) Depth-attention mechanism relies on depth maps with depth differences; therefore, the values of $D_\alpha$ and $D_\beta$ need to be taken more precisely when the depth difference is small.

**Author Contributions:** Conceptualization, J.Z. and J.L.; methodology, J.Z., J.L. and X.Q.; software, J.Z.; validation, B.L., Y.Z. and J.S.; formal analysis, H.L.; investigation, J.Z., X.Q., J.L., B.L., H.L., Z.L., J.S. and Y.Z.; resources, J.Z. and B.L.; data curation, J.Z., X.Q., J.L., B.L., H.L., Z.L., J.S. and Y.Z.; writing—original draft preparation, J.Z. and J.L.; writing—review and editing, J.Z., J.L. and X.Q. All authors have read and agreed to the published version of the manuscript.

**Funding:** This study was supported by the National Natural Science Foundation of China (grant nos. 62063030 and 62163032), the Financial Science and Technology Program of the XPCC (grant nos. 2021DB003, 2022CB011, and 2022CB002-07), and the High-Level Talent Project of Shihezi University (grant nos. RCZK2018C31 and RCZK2018C32).

**Institutional Review Board Statement:** Not applicable.

**Informed Consent Statement:** Not applicable.

**Data Availability Statement:** We published our dataset at https://github.com/zjlanthe/synthetic_data_insulator_blender, accessed on 7 September 2022.

**Conflicts of Interest:** The authors declare no conflict of interest.

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
