# Peer review of "A Fitting Recognition Approach Combining Depth-Attention YOLOv5 and Prior Synthetic Dataset"

_applsci, doi:10.3390/app122111122_

Round 1
Reviewer 1 Report
Dear Authors:
The article has been carefully reviewed.
The paper presents a fitting recognition approach combining depth-attention YOLOv5 and prior synthetic dataset to improve the validity of fitting recognition.
The proposed method has solved the problem of addressing power transmission lines (PTLs) traveling through complex environments leading to misdetections and omissions in fitting recognition using cameras.
The experimental part of the paper is divided into two environments for experimentation, and the results fully illustrate the effectiveness of the proposed method.
It is a topic of interest to the researchers in the related areas but I think, the paper needs some improvement for good understanding of the related researchers.
The reviewing comments are as follows:
1. In the paper, it is mentioned that the depth-attention network structure can be placed in the head of YOLOv5 or directly behind the trained network model, and it is suggested to add experiments that place the depth-attention network directly behind the trained network model for comparative illustration.
2. In the paper, it is mentioned that the training and testing run in ubuntu 18.04 LTS system, using a computer with graphics NVIDIA GTX-1660. It is recommended that this content be explain the detail process of the training and testing with features and characteristics of training-dataset.
3. The article mentioned that you introduce the ECA, CA, and SE attention mechanisms into the last layer and C3 module of the YOLOv5 backbone. However, in Figure 24, only the comparison chart of the results at position C3 is available, and it is recommended to add the comparison chart of the results at two positions.
4. In Figure 25, why is the time-costs of each group of synthetic data rising in the initial 4 seconds and only starting to fall in subsequent times? I just think, it is related with the features of training dataset and training process (it is related the above-mentioned No.2). If it is possible to check, the article should be better (if necessary).
5. In the 6.3 heading, there is an error in the writing of the parameter, please correct it.
6. Please double check the format of the references and incorrect reference formatting needs to be corrected.
I think that the subject of the paper is a good research for the aims of Special issues (Computer Vision, Robotics and Intelligent Systems) of the Journal.
Best regards,
Author Response
The response to the reviewer is in the annex.

Author Response

(The authors gave the same response as above.)
